# *Drosophila* COMPASS Complex Subunits Set1 and Ash2 Are Required for Oocyte Determination and Maintenance of the Synaptonemal Complex

**DOI:** 10.3390/jdb13030030

**Published:** 2025-08-19

**Authors:** Brigite Cabrita, Mary Enyioko, Rui Gonçalo Martinho

**Affiliations:** Department of Medical Sciences, Institute of Biomedicine (iBiMED), University of Aveiro, Agra do Crasto, Edifício 30, 3810-193 Aveiro, Portugal; b.cabrita@ua.pt (B.C.); mary.enyioko@ua.pt (M.E.)

**Keywords:** oocyte determination, synaptonemal complex, COMPASS complex, Set1, Ash2

## Abstract

Female gametogenesis is orchestrated by dynamic epigenetic modifications. In mammals, SETDB1, a histone H3K9 methyltransferase, is required for proper meiotic progression and early embryonic development. In *Drosophila*, the ortholog of SETDB1 plays a critical role in germ cell differentiation, transposon silencing, and the transcriptional repression of specific germline genes during oocyte fate determination. Moreover, Polycomb group (PcG) proteins in both mammals and *Drosophila* are essential for primary oocyte viability and meiosis, functioning through the silencing of early prophase I genes during later stages of prophase. While the repressive roles of epigenetic regulators in both *Drosophila* and mammalian oogenesis are well characterized, the functions of epigenetic activators remain less defined. Gene expression is controlled by the opposing activities of PcG and Trithorax group (TrxG) proteins, with the latter constituting a diverse family of chromatin remodelling factors that include H3K4 methyltransferases. In *Drosophila*, SET domain containing 1 (Set1)—the ortholog of mammalian SETD1A/B—acts as the primary regulator of global H3K4me2/3 levels. Set1 is critical for germline stem cell (GSC) self-renewal, functioning through both cell-autonomous and non-cell-autonomous mechanisms, with its depletion in the germline resulting in a progressive loss of GSC. More recently, Set1 has been implicated in germline cyst differentiation, although the mechanisms underlying this role remain poorly understood due to the complexity of the observed phenotypes. To investigate this, we analyzed ovaries from recently eclosed females in which Set1 and its highly conserved COMPASS partner, absent, small, or homeotic discs 2 (Ash2), were depleted—thus minimizing the confounding effects from GSC loss. We observed striking defects in both oocyte determination and Synaptonemal Complex (SC) integrity in one- to two-day-old females, within otherwise normal egg chambers. Interestingly, while defects in oocyte fate and oocyte–chromatin architecture were partially recovered in older egg chambers, SC integrity remained compromised. These findings suggest a critical window for SC assembly during germline cyst differentiation, after which this assembly cannot occur.

## 1. Introduction

Primary oocyte determination occurs in many organisms within a germline cyst, a multicellular structure composed of germ cells interconnected by cytoplasmic bridges [1]. Despite being in a cytoplasmic continuum, cells within each cyst can have dramatically distinct cell fates, such as becoming an oocyte versus supporting nurse cells. This is associated with symmetry breaking within the cyst and the asymmetric accumulation of cell fate determinants [2,3].

The structure and organization of the female germline cyst varies among metazoans, most likely to accommodate distinct developmental strategies of the oocyte. However, many of the epigenetic mechanisms supporting female gametogenesis are remarkably conserved [4,5,6,7,8,9]. For example, in mammals and *Drosophila*, oocyte determination during early prophase I occurs in the context of a transient weakening of Polycomb-mediated repression of early meiotic genes [10,11]. In contrast, oocyte differentiation during mid/late-prophase I and the establishment of the ovarian reserve depend on Polycomb-dependent repression of early prophase I genes [10,11,12,13]. Impairment of such Polycomb-mediated transcriptional repression is associated with meiotic defects, abnormal development and decreased viability of primary oocytes, and premature ovarian failure [11,12,13].

SET Domain Bifurcated Histone Lysine Methyltransferase 1 (SETDB1) is a methyltransferase responsible for the methylation of lysine 9 of histone 3 (H3K9). In mice, SETDB1 regulates the levels of histone H3 lysine 9 di-methyl (H3K9me2) and its loss in developing oocytes greatly impairs meiosis, with an arrest at the germinal vesicle and meiosis I stages [14,15]. Maternally deficient embryos for SETDB1 exhibit cell cycle defects and an arrest of development before implantation [14,15]. *Drosophila* SETDB1, also known as eggless, is similarly required for the deposition of repressive marks (e.g., H3K9me3) in the female germline [16,17]. Downregulation of *Drosophila* SETDB1 is important for female germline development, including germ cells differentiation during oogenesis and silencing of transposon and male-specific genes [16,17,18]. More recently, it was shown that similarly to Polycomb [10,13], *Drosophila* SETDB1 also contributes to the silencing of early prophase I genes upon oocyte specification [19]. Altogether, these results highlight the critical role of epigenetic repression of germline gene expression for correct determination, differentiation and maturation of *Drosophila* and mammalian oocytes.

While the role of epigenetic repression has been extensively described within the female germline, the role of epigenetic activators has been comparatively less explored. Gene expression is known to be tightly regulated by the opposing functions of the Polycomb Group (PcG) proteins and the Trithorax Group (TrxG) proteins [20,21]. TrxG is a family of proteins that form large protein complexes exhibiting histone methyltransferase and/or chromatin remodelling functions that positively regulate gene expression [22]. This family of proteins is named after the Trithorax (Trx) protein, a histone 3 lysine 4 (H3K4) methyltransferase [20]. However, Trx is not the only known H3K4 methyltransferase. In *Drosophila*, there are at least three partially redundant methyltransferases: SET domain containing 1 (Set1) (whose human orthologues are SETD1A and SETD1B), Trx (whose human orthologue is KMT2A), and Trithorax-related (Trr) (whose human orthologue is KMT2C) [20]. In mammals, there are at least six partially redundant H3K4 methyltransferases: SETD1A, SETD1B, MLL1/KMT2A, MLL2/KMT2B, MLL3/KMT2C, and MLL4/KMT2D. Among these, Set1 is the major H3K4 methyltransferase, as its loss in *Drosophila*, or the simultaneous loss of SETD1A and SETD1B in mammals, is associated with a major reduction in the bulk levels of di- and trimethylation of H3K4 [23,24].

Mammalian and *Drosophila* SET1 are components of a multimeric protein complex named COMPASS. SET1 interacts with a group of highly conserved regulatory proteins that are similarly important for regulating the bulk levels of H3K4me3 [23,25,26]. Deletion of mouse H3K4 methyltransferase SETD1B and its interacting protein CXXC1 produces similar defects during oogenesis, including an increase in atretic follicles, meiotic abnormalities, and zona pellucida defects [27]. Furthermore, upon fertilization, embryos fail to progress past the 2-cell stage [28]. SETD1B regulates H3K4me3 at promoters of actively expressed genes, including distinct transcriptional regulators expressed in the developing oocytes [27]. Deletion of SETD1B is also associated with a global redistribution of H3K4me3, showing reduced levels of DNA methylation and increased H3K4me3 in unmethylated CpG-rich DNA [27].

*Drosophila* Set1 is the major H3K4me3 methyltransferase, catalyzing the bulk of H3K4 di- and tri-methylation in the female germline [24,29]. Set1 is required cell-autonomously for the self-renewal of female germline stem cells (GSCs), as its germline depletion leads to a progressive loss of GSCs due to precocious differentiation [29]. This loss is likely due to an impairment of Dpp signalling within the GSCs [29]. Set1 is also required in the somatic stem cell niche for maintaining female GSCs; somatic depletion of Set1 non-cell-autonomously impairs GSC self-renewal due to Dpp signalling defects and reduced levels of DE-cadherin at the GSC-niche junctions [29].

*Drosophila* Set1 has also been reported to be required for differentiation of the germline cyst [30]. The defects in germline cyst differentiation after depletion of Set1 remain however poorly understood due to the complexity of the observed phenotypes. To overcome these difficulties, we focused in Set1 and its interacting COMPASS subunit Absent, small, or homeotic discs 2 (Ash2) [20,23,24], and we used recently eclosed females to minimize the confounding effects of GSC self-renewal defects after depletion of Set1.

We observed that females whose germline was depleted for Set1 or Ash2 formed morphologically normal-looking egg chambers, yet they exhibited striking defects in oocyte determination and Synaptonemal complex (SC) integrity during mid-pachytene. Interestingly, while defects in oocyte determination and oocyte–chromatin architecture partially recovered in older egg chambers, there was no detectable recovery of the SC assembly defects. This supports the hypotheses that there is a critical window during germline cyst differentiation for SC assembly or that the maintenance of SC integrity is particularly sensitive to Set1 and Ash2 depletion. Altogether, our results suggest that *Drosophila* Set1 and Ash2 are bona fide regulators of both oocyte determination and maintenance of SC integrity, possibly by positively regulating the timely expression of distinct early prophase I genes.

## 2. Materials and Methods

### 2.1. Drosophila Husbandry

*Drosophila melanogaster* flies were raised at 25 °C in standard medium (cornmeal, molasses, yeast, soya flour and beetroot syrup). All *Drosophila* crosses were performed at 25 °C where approximately 10–20 virgin females were, respectively, mated with 5–10 males. Flies were transferred to fresh vials after 48 h and occasionally after 72 h. For ovary dissection, F1 females were collected up to 1-day post-eclosion and transferred for 24 h to fresh vials containing standard medium supplemented with fresh yeast paste. All females under analysis were kept with males.

### 2.2. Germline-Specific RNAi

To specifically deplete SET domain containing 1 (Set1) and absent, small, or homeotic discs 2 (Ash2) within the female germline, the Gal4/UAS system and a germline-specific *nanos*-Gal4 (*nos*-Gal4-VP16) driver were used [31]. Males carrying UAS-RNAi were crossed to *nanos*-Gal4 virgin females. Females from the progeny (F1) were used for further analysis. As a negative control (control RNAi) it was used a short hairpin against mCherry, a gene that is not present in the *Drosophila* genome. A short hairpin against Kdm5 (Kdm5 RNAi) was used as a positive control for SC maintenance phenotype [32]. Germline-specific depletion of Set1 and Ash2 was achieved using short TRiP hairpins [33], which are available at Bloomington *Drosophila* Stock Center (BDSC). For consistency, we used the same TRiP hairpins and nanos-Gal4 driver as in previously published work [29,30]. For more information about used *Drosophila* stocks see Appendix A. For all experiments and controls, it was performed a minimum of two biological replicates.

### 2.3. Ovaries Immunostaining

Adult ovaries (10 to 20 ovary pairs per sample per experiment) were processed according to standard procedures. Briefly, ovaries were dissected from 1 to 2 days post eclosion females in ice cold phosphate-buffer saline (PBS) and fixed for 20 min in 4% paraformaldehyde, in PBS + 0.5% NP-40. After fixation samples were washed three times for 10 min each in PBST (0.2% Tween-20 in PBS). Following a gentle detachment, ovarioles were blocked for 2 h in PBST supplemented with 1% Triton X-100, 1% (*w*/*v*) bovine serum albumin (BSA) and 1% (*w*/*v*) donkey serum. Primary antibody incubation was performed overnight at 4 °C in PBST supplemented with 1% BSA and 1% donkey serum (BBT solution). After washing three times for 10 min each in PBST, ovaries were incubated for 4 h at room temperature with the appropriate secondary antibodies diluted in BBT solution. DNA was subsequently stained for 15 min at room temperature using 1:5000 DAPI in PBST. Prior to mounting, ovaries were washed two times with PBST, one time with PBS 1×. Mounting was performed in VECTASHIELD. Images were acquired using a Zeiss LSM880 Confocal microscope with a 63× Plan-Apo oil immersion objective (numerical aperture 1.4) (Carl Zeiss Microscopy GmbH, Jena, Germany), deconvolved using Huygens Remote Manager Software v3.9 (Scientific Volume Imaging B.V., Hilversum, The Netherlands), partially projected using Fiji Software (Version 1.54p), analyzed with Fiji or ZEN Blue software (Carl Zeiss Microscopy GmbH, Jena, Germany, ZEN 3.3 blue edition, version 3.3.89.0000 and ZEN 3.4 blue edition, version 3.4.91.0000). Primary and secondary antibodies used in this work, including working dilutions, sources, and references, are indicated in Appendix A.

### 2.4. Signal Analysis and Quantification

To quantify Synaptonemal complex (SC) maintenance and Orb enrichment, it was used a widefield upright microscope Zeiss AxioImager Z1 with a Plan-Apochromat 63×/1.40 Oil DIC objective (Carl Zeiss Microscopy GmbH, Jena, Germany). Images for Orb enrichment analysis with rainbow LUT were acquired using a Zeiss LSM880 Confocal microscope with a 63× Plan-Apo oil immersion objective (numerical aperture 1.4). Antibody immunostaining against Corolla was used as an SC marker. Two different monoclonal antibodies were used in immunostaining against Orb. DNA staining was used to stage egg chambers.

To quantify Orb levels, we employed two distinct methods: epifluorescence microscopy and confocal image analysis. For both approaches, Orb enrichment in the oocyte cytoplasm was classified in three categories: “Orb enrichment”, “Partial Orb enrichment” and “no Orb enrichment” (Figure 1A).

Epifluorescence analysis: “Orb enrichment” class was defined when Orb signal was strong and highly localized within the oocyte cytoplasm, being clearly distinguishable from the nurse cells. “Partial Orb enrichment” class referred to egg chambers showing a weaker Orb signal in the oocyte cytoplasm, accompanied by increased signal in the nurse cells. Egg chambers classified as having “no Orb enrichment” displayed a faint and diffuse Orb signal throughout the egg chamber, without clear oocyte localization.

For analysis of the confocal microscopy images, we applied a rainbow LUT to translate the Orb signal intensity into a colour scale (Figure 1B). For each z-stack, it was selected the section where the Orb signal was the highest. “Orb enrichment” class corresponded to egg chambers whose localized Orb signal was at least green or higher in the rainbow LUT colour scale. “Partial Orb enrichment” corresponded to egg chambers where localized Orb signal intensity was light blue. When the Orb signal was black or dark blue across the egg chamber without clear oocyte localization it was classified as having “no Orb enrichment”.

To quantify maintenance of SC integrity, we used three categories based on the SC signal in the oocyte nucleus: “Normal SC (Filamentous)”, “Dotted SC” and “no SC or unlocalized SC” (Figure 2B). Whenever a fully assembled, continuous SC structure was visible with no apparent breaks, we classified the SC as normal. In cases where the SC showed a disrupted pattern by a punctate (dotted) signal in the oocyte nucleus, we considered Dotted SC. When the SC signal was either completely absent from the oocyte nucleus or appeared diffuse throughout the oocyte nucleoplasm, we classified it as “No SC or Unlocalized SC”.

For assessment of karyosome morphology, we used three categories based on the z-stack projections of the oocyte DNA (DAPI staining): “Round shaped karyosome”, “Irregular shaped karyosome” and “Bigger karyosome (decondensed/polyploid)” (Figure 5A). “Round shaped karyosome” class was defined as a clearly visible, fully round structure of the oocyte nucleus. “Irregular shaped karyosome” class referred to oocyte nuclei with non-round morphology, but without a significant increase in size, compared to controls. “Bigger karyosome” class included oocyte karyosomes that were noticeably larger than controls.

### 2.5. Statistical Analysis

All statistical analyses were performed using GraphPad Prism 10.0 software. Contingency tables and Fisher’s exact test were used to analyze statistical differences between different groups. A 95% CI was used for all analyses. Detailed *p*-values between groups can be found in Appendix A.

## 3. Results

### 3.1. Set1 and Ash2 Are Required for Primary Oocyte Determination

Besides its cell-autonomous and non-cell-autonomous roles in regulating germline stem cell (GSC) self-renewal, Set1 has also been proposed to be required for germline cyst differentiation [30]. However, these differentiation defects remain difficult to interpret due to the complexity of the observed phenotypes. To address this, we focused on Set1 and its interacting COMPASS subunit, Ash2, and used recently eclosed females to minimize the confounding effects of GSC self-renewal defects. Unlike previous reports in older females, where germline depletion of Set1 was associated with the formation of highly abnormal pseudo-egg chambers [30], we observed in recently eclosed females that germline-specific depletion of Set1 or Ash2 resulted in morphologically normal-looking egg chambers. However, these chambers exhibited striking defects in oocyte determination and a significant loss of Synaptonemal Complex (SC) integrity by mid-pachytene.

Our initial observations indicate that both Set1 and Ash2 are required for oocyte determination. The oo18 RNA-binding protein gene (*orb*) encodes an RNA-binding protein essential for egg chamber formation and polarity establishment during oogenesis [34]. Orb is initially expressed throughout the 16-cell germline cyst, but it soon begins to progressively accumulate in the two pro-oocytes, and subsequently becomes restricted to a single cell, the future oocyte. As a result of the asymmetric accumulation of oocyte determinants within the germline cyst, one cell adopts the oocyte fate, while the remaining 15 differentiate as supporting nurse cells. As such, to assess oocyte fate determination, we evaluated Orb enrichment using two distinct experimental approaches: first, we used an epifluorescence microscope to score egg chambers in three distinct classes (Figure 1A); second, we analyzed representative confocal images using a ‘rainbow’ LUT to highlight Orb staining pixel intensity and objectively score egg chambers in three distinct classes (Figure 1B) (for more information see Materials and Methods (Section 2)).

We observed that Orb enrichment in the presumptive oocyte was impaired in region 3 of the germarium (mid-pachytene) following Set1 and Ash2 depletion (Figure 1A–C). Most egg chambers exhibited either no specific Orb enrichment or partial Orb enrichment in the oocyte cytoplasm (see region 3 of the germarium (R3) in Figure 1A–C). Interestingly, from mid- to late-pachytene (stages 2/3 and later), the frequency of egg chambers with Orb enrichment progressively increased compared to mid-pachytene (see stages 2/3 and later in Figure 1A–C), suggesting a partial recovery of the oocyte determination defects in later stages of oogenesis. These results were confirmed using two non-overlapping short-hairpin RNAs against Set1 (Set1 RNAi H1 and Set1 RNAi H2) (Figure 1 and Appendix A).

### 3.2. Set1 and Ash2 Are Required for Maintenance of SC Assembly

The Synaptonemal complex (SC) is a conserved protein structure that assembles between paired homologous chromosomes during meiosis, maintaining their pairing and alignment until meiotic recombination is completed. The 16-cell cyst differentiates into one oocyte and 15 supporting nurse cells. Interestingly, early on, four cyst cells enter meiosis and assemble an SC. Subsequently, three of these cells exit meiosis, disassemble the SC, and differentiate as supporting nurse cells, whereas the remaining cell maintains a fully assembled SC and becomes the oocyte [35].

Since maintenance of SC assembly during mid-pachytene occurs exclusively in the presumptive oocyte, we investigated whether this maintenance was impaired following Set1 and Ash2 depletion. To test this, we stained control, Set1-depleted and Ash2-depleted ovaries for the SC subunit Corolla. As a positive control for SC maintenance defects, we used *Drosophila* ovaries depleted for Lysine demethylase 5 (Kdm5) [32]. As expected, Kdm5 depletion resulted in a significant loss of SC integrity by mid-pachytene, visible as a fragmented SC (dotted SC) (Figure 2A,B). Depletion of Set1 and Ash2 also impaired maintenance of SC integrity during mid-pachytene (Figure 2A,B). Notably, the disassembly of the SC following Set1 and Ash2 depletion, but not following Kdm5 depletion, was associated with an apparent accumulation of the SC subunit Corolla in the oocyte nucleoplasm (Figure 2B).

The frequency of egg chambers with loss of SC integrity increased during mid/late-pachytene compared to earlier stages (compare regions 2/3 of germarium with stages 2/3 in Figure 2A), indicating that the loss of these chromatin remodelling proteins does not impair the initial assembly of the SC during early prophase I, but leads to a progressive loss of SC integrity during mid-to-late pachytene. Overall, like Kdm5, Set1 and Ash2 are similarly required for maintaining SC assembly during mid-to-late pachytene.

### 3.3. Delayed Oocyte Determination Fails to Rescue SC Assembly

Our results indicate that Set1 and Ash2 are required for oocyte determination and maintenance of SC assembly. We wondered, however, whether the partial recovery of Orb enrichment in the oocyte during mid/late pachytene was also associated with an improvement in SC integrity. To investigate this, we co-stained Set1- and Ash2-depleted ovaries for both Orb and Corolla and examined the correlation between Orb enrichment and SC maintenance.

Control egg chambers (mCherry RNAi) invariably showed a normal SC (filamentous) during pachytene of prophase I (Figure 3A,B), even when there was only a partial enrichment of Orb in the oocyte (Figure 3A). As expected [32], depletion of Kdm5 (Kdm5 RNAi) was associated with a loss of the SC integrity (dotted SC) during mid/late-pachytene (stages 2/3) independently of oocyte determination defects (Orb oocyte-enrichment) (Figure 3A,B). Depletion of Set1 and Ash2 was associated with loss of SC integrity during mid and mid/late-pachytene (region 3 and stages 2/3). However, the increased frequency of egg chambers with normal Orb enrichment during mid/late pachytene (stage 2/3) did not correlate with any detectable recovery of SC assembly defects (Figure 3A,B).

Overall, our observations suggest that while SC maintenance defects following Kdm5 depletion occur largely independently of oocyte determination loss, the SC defects observed after Set1 and Ash2 depletion correlate with early defects in oocyte determination. Interestingly, the partial recovery of oocyte determination during mid/late pachytene (stage 2/3) did not lead to a detectable rescue of SC assembly defects.

### 3.4. Set1 and Ash2 Are Required for Germline H3K4 Trimethylation

H3K4 trimethylation (H3K4me3) is a histone modification typically associated with active gene transcription and is enriched at gene promoters. H3K4me3 levels were significantly higher in the transcriptionally active nurse cells compared to the oocyte (see control RNAi in Figure 4). *Drosophila* Set1 is the major H3K4me3 methyltransferase, catalyzing the bulk of H3K4 di- and tri-methylation in the female germline [24,29]. Consistently, the levels of H3K4me3 were significantly reduced in the germline following Set1 and Ash2 depletion (Figure 4).

Overall, our observations further support a key role for Set1, and its interacting partner Ash2, in regulating H3K4me3 levels within the germline, and confirm the effectiveness of our knockdown experiments.

### 3.5. Delayed Karyosome Formation Does Not Correlate with Recovery of SC Assembly

Once the meiotic double-strand breaks are correctly repaired during mid-pachytene, thereby satisfying the DNA damage checkpoints [36,37], the oocyte–chromatin detaches from the nuclear envelope and reorganizes into a compacted structure called the karyosome [38]. This reorganization of the chromatin is a distinctive feature of the *Drosophila* oocyte and overlaps with a global repression of transcription [6,39].

We decided to examine karyosome formation to further investigate the partial recovery of oocyte fate following Set1 and Ash2 depletion. Previous studies indicate that depletion of Kdm5 alters karyosome morphology during mid/late prophase I [6,32]. In our experiments, depletion of Set1 or Ash2 similarly resulted in altered karyosome morphology during early stages (stages 3/4 in Figure 5A). However, unlike Kdm5 depletion, we observed a partial recovery of karyosome architecture during late pachytene/early diplotene (stages 5/6 in Figure 5A).

The recovery of karyosome formation by mid/late prophase I supports our working hypothesis that, at least in our experimental setup using young females and the nanos-Gal4 driver, germline depletion of Set1 and Ash2 is associated with a partial recovery of oocyte-like characteristics at later stages of oogenesis. However, SC integrity remained compromised, indicating that delayed oocyte determination did not rescue this defect. Consistently, there was no correlation between SC integrity and the recovery of karyosome morphology in later stages of oogenesis (Figure 5B).

## 4. Discussion

Besides its role in germline stem cell (GSC) maintenance, *Drosophila* Set1 was also reported to be required for the differentiation of the germline cyst [30]. However, this differentiation defects remain poorly understood due to the complexity of the observed phenotypes. To overcome these difficulties, we used recently eclosed females to minimize the confounding effects of GSC self-renewal defects.

We observed that females one- to two-days post-eclosion, with germlines depleted of Set1 or Ash2, formed morphologically normal-looking egg chambers. However, they exhibited defects in oocyte determination and a loss of the Synaptonemal complex (SC) integrity during mid-pachytene. Set1 is an H3K4 methyltransferase whose function is critical for establishing the correct levels of H3K4me2 and H3K4me3 at the promoters of actively transcribed germline genes [30]. We therefore propose that the Set1/COMPASS complex positively regulates the expression of a subset of germline genes required for *Drosophila* primary oocyte determination and the maintenance of SC assembly (Figure 6). Among these, orb is a germline gene potentially regulated by Set1 and Ash2. Orb has been shown to display high levels of H3K4me3 during germline cyst differentiation, and its reduced expression has been associated with delayed oocyte determination [40,41].

Although major defects in oocyte determination and SC integrity were evident by mid-pachytene, oocyte determination and oocyte–chromatin architecture partially recovered during late pachytene, while the SC assembly defects persisted. The partial recovery of oocyte determination defects might be attributed to the oocyte-determinant Orb stimulating its own translation via binding to its 3’UTR [40], or could simply result from the expression of the germline-specific *nanos*-Gal4 driver [31], whose ability to drive germline transcription is transiently reduced during mid- to late-pachytene [42]. Nonetheless, our results support the hypothesis that there is a critical window during germline cyst differentiation for SC assembly, after which this assembly cannot occur, even if other oocyte features are recovered. Alternatively, this might result from maintenance of SC integrity being especially vulnerable to Set1 and Ash2 depletion.

We hypothesize that after the loss of SC integrity during mid-pachytene due to the depletion of Set1 or Ash2, the germline cyst cells by mid-to-late-prophase I are not competent anymore to reassemble the SC, even after the delayed accumulation of oocyte-fate determinants (e.g., Orb) and the condensation of the oocyte–chromatin into a karyosome. The apparent irreversibility of SC disassembly might be the result of the inability to pair the homologous chromosomes at this stage of development or due to the fact that the germline cyst cells have accumulated several chromatin remodelling proteins known to downregulate the expression of several early prophase I genes, including SC components [10,13,19,43].

Altogether, our results indicate that Set1 and Ash2, along with other positive regulators of gene expression, contribute to the correct timing of oocyte determination within the germline cyst. Although causality has not been established, the timing of oocyte determination is likely crucial for maintaining SC integrity during mid-pachytene.

## 5. Final Remarks and Conclusions

The role of epigenetic activators in *Drosophila* germline cyst differentiation is not yet fully understood. We aimed to explore the role of COMPASS subunits Set1 and Ash2 in recently eclosed females. Upon germline-specific depletion of Set1 and Ash2, we observed defects in oocyte determination, karyosome formation, and SC maintenance within otherwise normal-looking egg chambers. This supports the role of Set1 and Ash2 in oocyte determination and differentiation, likely due to an abnormal expression of distinct early prophase I genes critical for primary oocyte formation.

Interestingly, we also observed a partial recovery of some, but not all, oocyte phenotypes during mid-to-late pachytene. While defects in oocyte determination and chromatin architecture were partially rescued at later stages of oogenesis, the premature disassembly of the SC during early- to mid-pachytene following Set1 and Ash2 depletion remained irreversible. We propose that while certain aspects of primary oocyte identity can be potentially restored later in oogenesis, others—such as SC integrity—are likely to be irreversibly lost if oocyte determination is delayed.

## Figures and Tables

**Figure 1 jdb-13-00030-f001:**
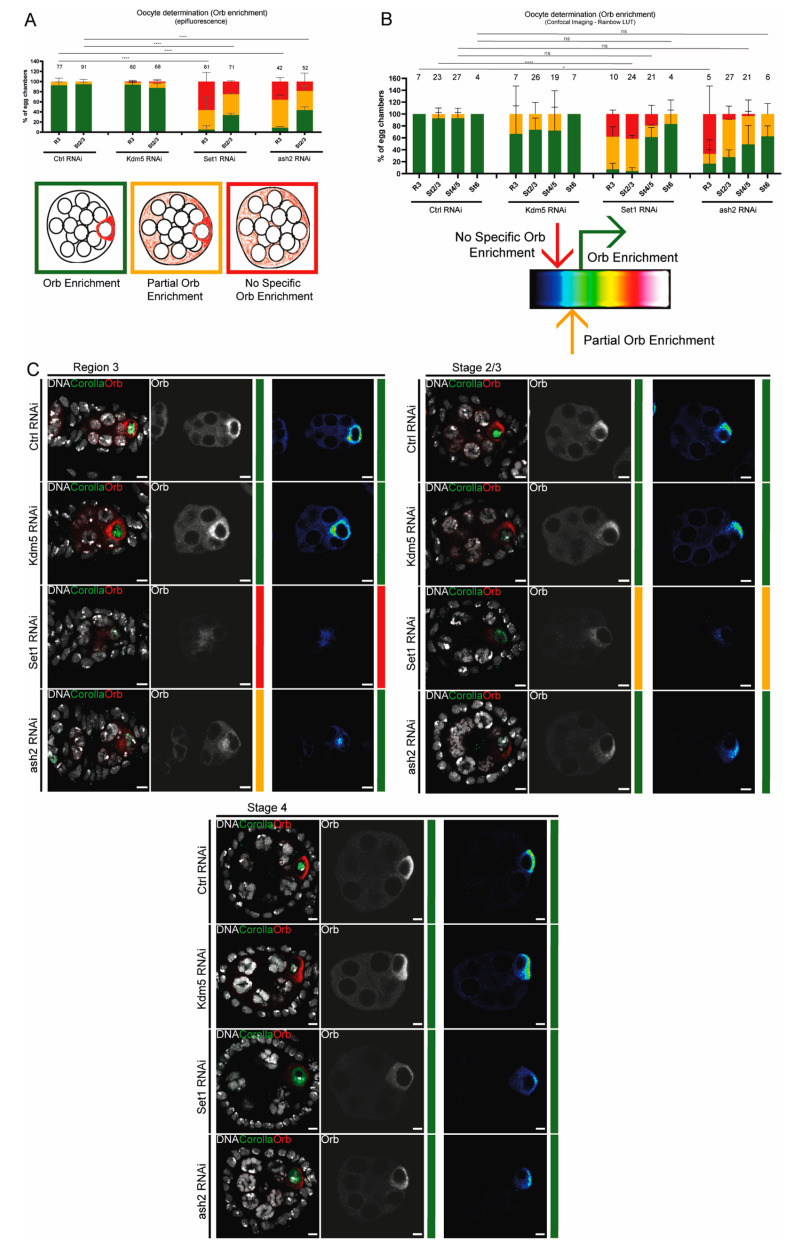
Set1 and Ash2 are required for primary oocyte determination. (**A**) Quantification of Orb enrichment by epifluorescence for control (mCherry RNAi), Kdm5 RNAi, Set1 RNAi and ash2 RNAi. The number of scored egg chambers is indicated at the top of each bar. Drawings depict the categories used to evaluate Orb enrichment and indicate the corresponding graph colour. Results are presented as mean ± SD. **** *p* < 0.0001. (**B**) Quantification of Orb enrichment by confocal imaging using LUT rainbow for control (mCherry RNAi), Kdm5 RNAi, Set1 RNAi and ash2 RNAi. The number of scored egg chambers is indicated at the top of each bar. Drawing indicates a representation of the LUT rainbow used and the criteria used to score each category. Results are presented as mean ± SD. * *p* < 0.05, **** *p* < 0.0001 and ns – non-significant. (**C**) Immunofluorescence images representative of Orb enrichment from mid to late pachytene for control (mCherry RNAi), Kdm5 RNAi, Set1 RNAi and ash2 RNAi. Overlay images show the egg chambers for each stage, with Orb being shown in a single channel in grey, followed by the rainbow LUT image. Scale bar: 4.7 μm. Overlay and Orb single channel in grey are partial z-projections. Rainbow LUT images show the z-step with the highest intensity of Orb signal. Ctrl—Control; R3—region 3 of the germarium; St2/3—Stages 2 and 3.

**Figure 2 jdb-13-00030-f002:**
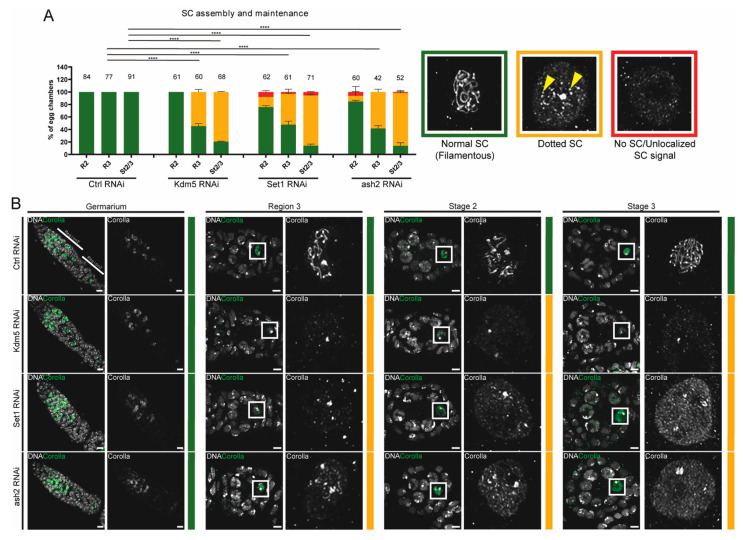
Set1 and Ash2 are required for maintenance of SC assembly. (**A**) Quantification of SC assembly for control (mCherry RNAi), Kdm5 RNAi, Set1 RNAi and ash2 RNAi. The number of scored egg chambers is indicated at the top of each bar. Representative images showing the Corolla signal for each category used to score SC integrity are provided with the corresponding graph colour. Results are presented as mean ± SD. **** *p* < 0.0001. (**B**) Immunofluorescence for Corolla in control (mCherry RNAi), Kdm5 RNAi, Set1 RNAi and ash2 RNAi. From left to right: DNA (grey) and Corolla (green) are shown in overlay. Corolla single channel (grey) is showing the inset marked in the white square. Scale bar: 4.7 µm. Images are partial z-projections of each region of the germaria or egg chamber. Ctrl—Control; R3—Region 3 of the germarium; St2/3—Stages 2 and 3 of oogenesis.

**Figure 3 jdb-13-00030-f003:**
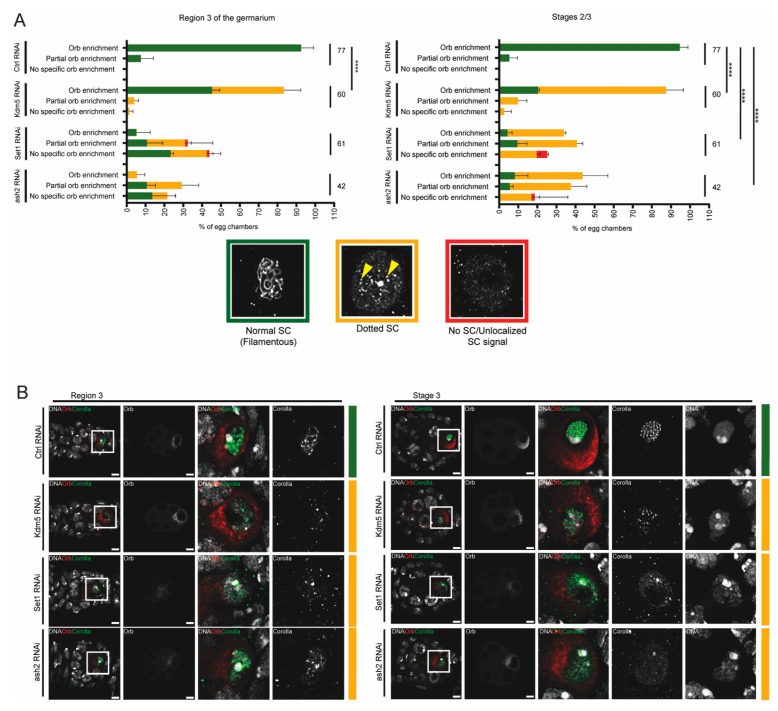
Delayed determination of the oocyte fails to rescue SC assembly. (**A**) Correlation between Orb enrichment and SC integrity during mid-pachytene (region 3) and mid-to-late of prophase I (Stages 2/3). For each egg chamber, Orb enrichment was analyzed first, followed by assessment of SC integrity. Representative images showing the Corolla signal for each category used to score SC integrity are provided with the corresponding graph colour. The number of scored egg chambers is indicated for each condition. Results are presented as mean ± SD. **** *p* < 0.0001. (**B**) Immunofluorescence for Orb (oocyte marker) and Corolla (SC subunit) in Control RNAi, Kdm5 RNAi, Set1 RNAi and ash2 RNAi ovaries. Scale bar: 4.7 μm. Images are partial z-projections of each region/egg chamber. Ctrl—Control.

**Figure 4 jdb-13-00030-f004:**
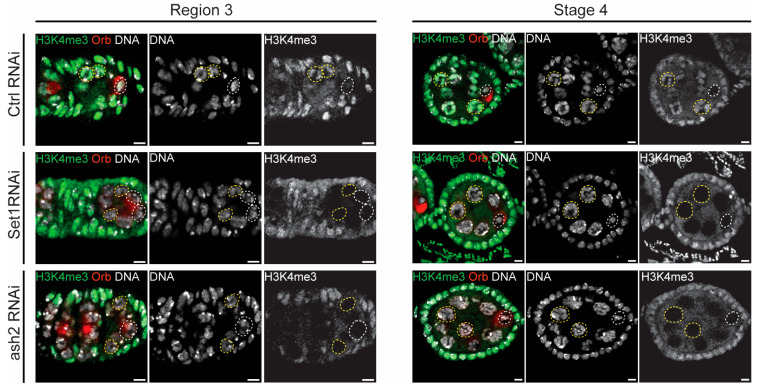
Set1 and Ash2 are required for germline H3K4 trimethylation (H3K4me3). Immunofluorescence for H3K4me3 and Orb (oocyte marker) in Control (mCherry RNAi), Set1 RNAi and Ash2 RNAi in early pachytene (region 3) and late pachytene (Stage 4). From left to right: DNA (grey), H3K4me3 (green) and Orb (red) are shown in overlay; DAPI and H3K4me3 are shown in single channel in separate images. Yellow dashed lines outline the nurse cells, while white dashed lines indicate the oocyte nucleus. Scale bar: 4.7 µm. Images are partial z-projections of each region 3 or Stage 4. Ctrl—Control.

**Figure 5 jdb-13-00030-f005:**
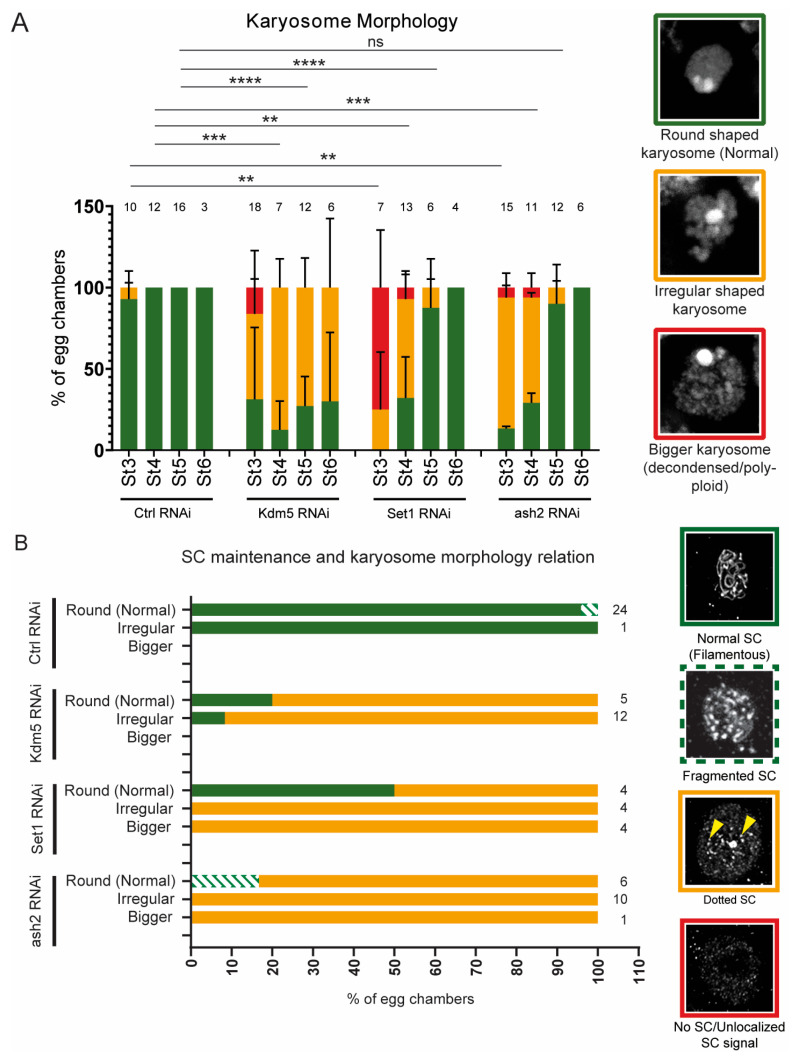
Delayed karyosome formation does not correlate with a recovery of SC assembly. (**A**) Quantification by confocal imaging of karyosome morphology of Control RNAi, Kdm5 RNAi, Set1 RNAi and ash2 RNAi. The number of scored egg chambers is indicated at the top of each bar. DNA images representative of each karyosome category are shown with the corresponding graph colour. Results are presented as mean ± SD. ** *p* < 0.005, *** *p* < 0.0005, **** *p* < 0.0001 and ns—non-significant. (**B**) Correlation between SC assembly and karyosome morphology in late pachytene (stages 3 to 5 of *Drosophila* oogenesis). The number of scored egg chambers is indicated at the top of each bar. For each stage, we first assessed the karyosome morphology followed by the analysis of SC assembly. Representative images showing the Corolla signal for each category used to score SC integrity are shown with the corresponding graph colour.

**Figure 6 jdb-13-00030-f006:**
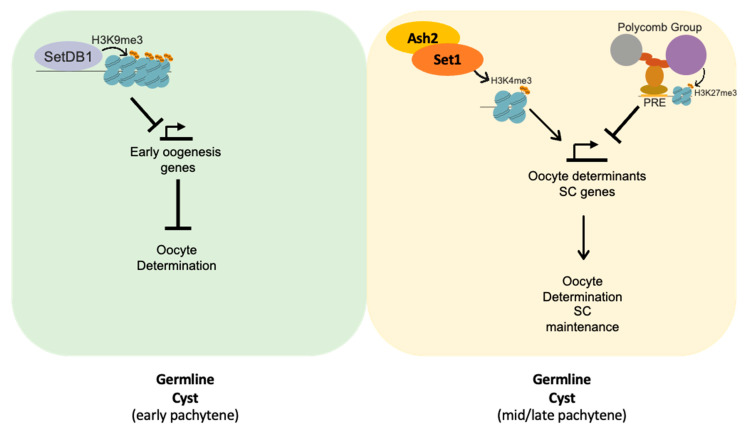
Working model: Set1 and Ash2 are required for expression of a subset of genes required for oocyte fate determination and maintenance of SC integrity. In early pachytene, SetDB1 is required to negatively regulate the expression of a subset of early oogenesis genes, promoting oocyte fate determination [19]. As the germline cyst progresses to mid-pachytene, the activity of the Polycomb group (PcG) proteins increases to negatively regulate the expression of early prophase I genes [10,13]. As such, we hypothesize that Set1 and Ash2 are required to counteract PcG-mediated repression and promote the correct expression of oocyte determinants, such as orb, as well as SC components.

## Data Availability

The data presented in this publication are available upon request from the corresponding author.

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
