# Peer review of "Drosophila COMPASS Complex Subunits Set1 and Ash2 Are Required for Oocyte Determination and Maintenance of the Synaptonemal Complex"

_jdb, 2025, doi:10.3390/jdb13030030_

Round 1

Reviewer 1 Report

Comments and Suggestions for Authors
  1. Scale bars in some Figures are are either not shown or are not displayed clearly.
  2. Based on the hypothesis described in the Disccusion part, could the authors draw a diagram to facilitate a better understanding of the hypothesis?

Reviewer 2 Report

Comments and Suggestions for Authors

This study investigates the role of the COMPASS complex subunits Set1 and Ash2 in Drosophila oogenesis, specifically in oocyte determination and synaptonemal complex maintenance. Using germline-specific RNAi in recently eclosed females, the authors show that depletion of Set1 or Ash2 leads to defective oocyte fate determination and premature disassembly of the SC during mid-pachytene, despite forming morphologically normal egg chambers. The study highlights that epigenetic activation, not just repression, is essential for female gametogenesis and meiotic progression in Drosophila. Overall, this study is well designed and has some novelty. I have several comments about it:

  1. The rationale of using Set1 and Ash2 in this study is not clear. Why do authors use Ash2, rather than other common subunit of COMPASS complex, such as wds and rbbp5, or the unique subunit, such as Hcf and Wdr82.
  2. The resolution for Figure 1 is too low. The author have to improve the quality of it.
  3. H3K4 methylation level also needs to be checked in Set1 and Ash2 knockdown animals.

Reviewer 3 Report

Comments and Suggestions for Authors

The manuscript by Cabrita and Martinho describes a role for the Set1 and Ash2 chromatin regulators in female germline development. The manuscript is well written, and the data provide strong evidence, albeit descriptive, for these genes in several phenotypic observations of germline developmental defects. Specifically, the authors demonstrate that germline-specific knockdown of Set1 or Ash2 lead to defects in Orb localization, SC structural integrity, and karyosome morphology. The results should be of interest to the readership of JDB. I have a few comments and suggestions for improvement. Following the authors’ addressing of these issues, publication is recommended.

Major comments:

  • Perhaps it is my copy of the draft manuscript, but the figures appear quite low resolution. This includes not only the cell/tissue images, but also the bar graphs. For example, Figure 1A and B graphs are difficult to read due to resolution of the text. If possible, the authors should render higher resolution graphics.

  • The suggestion of a partial recovery of oocyte determination defects in later stages (Lines 221-222) should be more cautiously interpreted. Although this is a possibility, it is also possible that the RNAi effects wane in later stages (in fact, the authors allude to this later in Lines 338-339). Perhaps the Orb phenotype is simply not as robust to the RNAi effects compared to the SC phenotype. This is important in relation to further interpretations described in Lines 285-288, where the authors suggest a causative link between oocyte determination defects and SC defects. Again, if the SC morphology is simply more sensitive to Set1/Ash2 knockdown, it is possible their defects remain even when the RNAi efficiency may be reduced in later developmental stages that allow for a partial recovery in the Orb phenotype.
    • Did the authors check for RNAi knockdown efficiency of Set1 or Ash2? If Set1 or Ash2 antibodies are reasonably accessible/available, the authors should stain and image for these. This would (1) confirm normal Set1 and Ash2 expression the germline, (2) assess the RNAi efficiency at the protein level by quantifying the loss of their expression, and (3) also determine if Set1 or Ash2 expression increases in later stages that could explain reversal of Orb phenotype.

Minor comments:

  • Check the phrase “Loss of Drosophila SETDB1 is important…” on Line 61. Is it correct that “loss of” the SETDB1 is important, or is this an error? If it is correct, I would suggest rephrasing “Loss of” to “Inactivation of” or something equivalent (“Downregulation of” etc.), as the term “Loss of” could be confused with a genetic manipulation rather than a natural reduction of the gene’s expression and/or activity.

  • The authors should provide a short (1 or 2 sentence) description of the Synaptonemal complex (SC) following its first use in the text (Line 112).

  • Similarly, on Line 194 when “Orb enrichment” is first introduced as a phenotype, the authors should provide a more specific description of what this means, why it is important, and more details for how the three phenotypic categories were scored (this latter aspect could be included as additional details in the Materials and Methods).

  • As with the categorical Orb enrichment phenotype assessment, the Materials and Methods would be improved with additional details for what specific and distinctive features were used to determine the 3 different SC phenotypes scored, as well as with the 3 Karyosome morphology phenotypes (which are potentially even less obvious to a non-expert reader simply from the images provided in Figure 4A).

  • The phrase “was associated to” on Line 270 is confusing. I would suggest changing to “was associated with” or perhaps “resulted in”.

  • In the Discussion, Lines 330-333, can the authors speculate on possible candidate genes that could provide a mechanistic model for how Set1 and Ash2 function? Are there known genes impacted by Set1/Ash2 that are linked to germline development or other general stem cell or developmental processes that the authors could expand on here?

Reviewer 4 Report

Comments and Suggestions for Authors

In manuscript jdb-3611066 the authors report defects in the female germ line of Drosophila upon Set1 and Ash2 depletion through RNAi, defects being mainly localized in the synaptonemal complex and in the oocyte determination. Overall, the manuscript is sound, results are convincing and the experimental approach adequate for the aim of the study, thus I do not see any major issues in the scientific setup. My main suggestion is to improve the quality of the images, that in my pdf copy were of bad quality, especially for the low resolution of the writings over the bars (Figure 1A is barely readable, figure 1B still pixelated, and writings are of different size compared to 1A, all other pictures suffer from low resolution of all writings especially on micrographs); also, size bars are barely visible in all pictures, consider making them thicker. Curiously, supplemental pictures have higher resolution and readability. “Melanogaster” in line 122 should be italicized.

Round 2

Reviewer 2 Report

Comments and Suggestions for Authors

The authors fully addressed my comments, and I don’t have any questions.

Reviewer 3 Report

Comments and Suggestions for Authors

The authors have resolved all of my comments and suggestions. I recommend publication at this point.